# Predictive Modeling of Phenotypic Antimicrobial Susceptibility of Selected Beta-Lactam Antimicrobials from Beta-Lactamase Resistance Genes

**DOI:** 10.3390/antibiotics13030224

**Published:** 2024-02-28

**Authors:** Md. Kaisar Rahman, Ryan B. Williams, Samuel Ajulo, Gizem Levent, Guy H. Loneragan, Babafela Awosile

**Affiliations:** School of Veterinary Medicine, Texas Tech University, Amarillo, TX 79106, USA; kaisar.rahman@ttu.edu (M.K.R.); ryan.b.williams@ttu.edu (R.B.W.); sajulo@ttu.edu (S.A.); gizem.levent@ttu.edu (G.L.); guy.loneragan@ttu.edu (G.H.L.)

**Keywords:** predictive modeling, beta-lactamase gene, MIC, Enterobacteriaceae

## Abstract

The outcome of bacterial infection management relies on prompt diagnosis and effective treatment, but conventional antimicrobial susceptibility testing can be slow and labor-intensive. Therefore, this study aims to predict phenotypic antimicrobial susceptibility of selected beta-lactam antimicrobials in the bacteria of the family Enterobacteriaceae from different beta-lactamase resistance genotypes. Using human datasets extracted from the Antimicrobial Testing Leadership and Surveillance (ATLAS) program conducted by Pfizer and retail meat datasets from the National Antimicrobial Resistance Monitoring System for Enteric Bacteria (NARMS), we used a robust or weighted least square multivariable linear regression modeling framework to explore the relationship between antimicrobial susceptibility data of beta-lactam antimicrobials and different types of beta-lactamase resistance genes. In humans, in the presence of the *bla*_CTX-M-1_, *bla*_CTX-M-2_, *bla*_CTX-M-8/25_, and *bla*_CTX-M-9_ groups, MICs of cephalosporins significantly increased by values between 0.34–3.07 μg/mL, however, the MICs of carbapenem significantly decreased by values between 0.81–0.87 μg/mL. In the presence of carbapenemase genes (*bla*_KPC_, *bla*_NDM_, *bla*_IMP_, and *bla*_VIM_), the MICs of cephalosporin antimicrobials significantly increased by values between 1.06–5.77 μg/mL, while the MICs of carbapenem antimicrobials significantly increased by values between 5.39–67.38 μg/mL. In retail meat, MIC of ceftriaxone increased significantly in the presence of *bla*_CMY-2_, *bla*_CTX-M-1_, *bla*_CTX-M-55_, *bla*_CTX-M-65_, and *bla*_SHV-2_ by 55.16 μg/mL, 222.70 μg/mL, 250.81 μg/mL, 204.89 μg/mL, and 31.51 μg/mL respectively. MIC of cefoxitin increased significantly in the presence of *bla*_CTX-M-65_ and *bla*_TEM-1_ by 1.57 μg/mL and 1.04 μg/mL respectively. In the presence of *bla*_CMY-2_, MIC of cefoxitin increased by an average of 8.66 μg/mL over 17 years. Compared to *E. coli* isolates, MIC of cefoxitin in *Salmonella enterica* isolates decreased significantly by 0.67 μg/mL. On the other hand, MIC of ceftiofur increased in the presence of *bla*_CTX-M-1_, *bla*_CTX-M-65_, *bla*_SHV-2_, and *bla*_TEM-1_ by 8.82 μg/mL, 9.11 μg/mL, 8.18 μg/mL, and 1.04 μg/mL respectively. In the presence of *bla*_CMY-2_, MIC of ceftiofur increased by an average of 10.20 μg/mL over 14 years. The ability to predict antimicrobial susceptibility of beta-lactam antimicrobials directly from beta-lactamase resistance genes may help reduce the reliance on routine phenotypic testing with higher turnaround times in diagnostic, therapeutic, and surveillance of antimicrobial-resistant bacteria of the family Enterobacteriaceae.

## 1. Introduction

Antimicrobial resistance (AMR) has risen to become a significant global concern, driven by the rapid increase in AMR infection rates [1]. AMR develops and disseminates in microorganisms in response to stressful environmental signals, usually in the presence of antimicrobial selection pressure, which eventually leads to reduced efficacy of antimicrobial therapy [2,3]. In the United States alone, antimicrobial resistance (AMR) is associated with 2.8 million infections and more than 30,000 deaths, with an extra USD 20 billion in healthcare costs annually [4,5]. Global estimates indicate that deaths directly attributed to AMR exceeded 1.2 million in 2019, with a projected increase to around 10 million annually by 2050 if no additional measures are taken to combat AMR [6].

An important driver of AMR in the human healthcare setting is the injudicious use and misuse of antimicrobials. These improprieties are primarily due to non-prescription uses, improper prescription, and poor regulation, of antimicrobials [7,8]. According to the World Health Organization (WHO), certain antimicrobials—such as the third and fourth-generation cephalosporins and carbapenems—are classified as critically important due to being the last resort for human treatment against antimicrobial-resistant bacteria [9,10]. It is, therefore, of utmost importance to protect the efficacy of these antimicrobials. However, in recent years, there have been increasing reports of resistance even to these critically important antimicrobials. Part of the strategic objectives of the World Health Organization’s (WHO) Global Action Plan (GAP) is to optimize the use of antimicrobials by improved diagnostics and strengthen knowledge of AMR through surveillance and research [11].

The outcome of the clinical management of antimicrobial-resistant infection is dependent on both proper and timely diagnosis and treatment [12]. Clinical diagnosis of antimicrobial-resistant pathogens has been largely identified via phenotype-based methods such as disk diffusion and minimum inhibitory concentration (MIC). These methods offer high reliability and accuracy and represent the gold standard as compared to other diagnostics procedures; however, the procedures are time-consuming and labor-intensive [13,14,15]. While accuracy in diagnosis is important for the treatment of microbial infection when antimicrobial-resistant pathogens are present, the timeliness of diagnosis may be equally critical. For example, delayed treatment of bloodstream infection (BSI), one of the most fatal antimicrobial-resistant infections and a leading cause of death globally with around two million in North America and one-quarter of a million in Europe, worsen clinical outcomes and increases the risk of mortality [12,14,16,17]. While efforts have been made to expedite AMR diagnoses through the adoption of molecular tools with proven results, a challenge associated with these is that the AMR genotype does not always correspond to the AMR phenotype due to silent genes, environmental influences on gene expression, and the complex interplay of multiple genetic elements, resulting in clinical treatment failure [18]. Data-driven approaches have shown promise in providing effective prediction of AMR phenotype from genotype to aid effective clinical management [19,20,21,22], however, there is still a shortage of scalable data-driven analytical approaches suitable for clinical use [22]. 

Clinical diagnosis of AMR depends on phenotypic antimicrobial susceptibility tests with a turnaround time of at least 48 hours [23]. Meanwhile, molecular methods can detect pathogens and resistance genes within a day [24]. One increasingly utilized method for predicting AMR outside of standard phenotypic testing for susceptibility is machine learning [25,26]. However, this approach relies on predictive models resulting from data that are highly specific to a geographic location or a single pathogen and are therefore not generalizable [27]. Although recently, machine learning models have incorporated more robust global datasets to overcome this challenge [20]. For an improved ability to monitor AMR at the global level and simultaneously aid in the identification, and treatment of AMR phenotypes, a data-driven method utilizing heterogeneous geographies and phenotypes is needed. 

The use of a linear regression model for analysis of existing surveillance data, which contains both AMR phenotypes and genotypes, has the potential to offer reliable predictions of susceptibility or resistance from important genes or AMR determinants, which can provide additional support and/or replacement of the existing antimicrobial susceptibility testing [28]. The use of such predictive models may offer a reduction in susceptibility testing turnaround time, support for data-driven and evidence-based empirical antimicrobial selection, and support for efficient and efficacious antibiotic dosing regimens–especially in cases when there are no established antimicrobial susceptibility breakpoints. A predictive model that incorporates robust parameters could also improve decision-making in the use of AMR genotype data as a proxy for characterizing AMR phenotypes in surveillance of AMR in humans, animals, and the environment.

We posit that the use of data that account for multiple variables, and phenotype patterns could easily and reliably be predicted from genotypes and offer better predictive model generalizability. Based on Pfizer’s Antimicrobial Testing Leadership and Surveillance (ATLAS) human datasets, we aim to develop a predictive model to estimate the phenotypic antimicrobial susceptibility of critical and high-priority antibiotics from AMR genotype data while selecting for various factors, such as organism types (Gram-positive and Gram-negative), infection types (Intra-abdominal, lower respiratory, urinary tract, bloodstream, skin and community-acquired and hospital-acquired), and spatiotemporal features (country and year). In addition, we use National Antimicrobial Resistance Monitoring System (NARMS) for Enteric Bacteria retail meat data to predict the minimum inhibitory concentration (MIC) of cephalosporin antimicrobials from beta-lactamase genes with other selected variables such as sampling year, bacterial genus (*Escherichia coli* and *Salmonella enterica*) and meat types (chicken breast, ground beef, ground turkey, and pork cuts). Prediction of phenotypic antimicrobial susceptibility information directly from genotype data will help reduce the reliance on routine phenotypic testing, thereby resulting in more timely diagnoses, quicker therapeutics, and improved clinical outcomes of antimicrobial-resistant infections.

## 2. Results

### 2.1. Prediction of MIC: Human Dataset from ATLAS

#### 2.1.1. Cefepime

Robust model was used in the cefipime model due to heteroscedasticity. From the model, cefepime’s minimum inhibitory concentration (MIC) exhibited significantly higher values in several countries, namely China, Croatia, Greece, India, Italy, Kenya, Latvia, Nigeria, Poland, Romania, Taiwan, and Ukraine. Conversely, it was significantly lower in Australia, Austria, Belgium, Canada, Colombia, Denmark, France, Germany, Hong Kong, Ireland, Japan, Netherlands, New Zealand, Philippines, Portugal, Slovenia, Spain, Sweden, and the United Kingdom compared to Argentina. When comparing it to *E. coli*, the MIC of *Citrobacter* spp., *Enterobacter* spp., and *Providencia* spp. displayed significantly higher values, while the MIC of *Klebsiella* spp. and *Serratia* spp. demonstrated significantly lower values. Regarding age groups, the MIC was higher (1.11 μg/mL) in the 65-to-84-years age range. In medical ICU settings, the MIC of cefepime was higher compared to clinical settings. Furthermore, for every one-year increase, the MIC of cefepime increased by 1.05 μg/mL. In the presence of the *bla*_CTX-M-1_, *bla*_CTX-M-2_, *bla*_CTX-M-8/25_, and *bla*_CTX-M-9_ groups of beta-lactamase genes, the likelihood of cefepime’s MIC was significantly increased by 6.98 μg/mL, 5.46 μg/mL, 4.41 μg/mL, and 3.98 μg/mL, respectively. Similarly, the MIC of cefepime was raised at 3.24 μg/mL, and 2.98 μg/mL more likely in the presence of *bla*_VEB_ and *bla*_PER_ genes. Moreover, in the presence of carbapenemase genes *bla*_NDM_, *bla*_VIM,_ and *bla*_IMP_, the MIC of cefepime was significantly increased by 2.09 μg/mL, 3.28 μg/mL, and 2.09 μg/mL, respectively. On the other hand, in the presence of *bla*_CMY-2_ and *bla*_DHA_, the MIC of cefepime was significantly decreased by 0.37 μg/mL and 0.33 μg/mL respectively. (Table 1).

#### 2.1.2. Ceftazidime

A robust model was used in the ceftazidime model due to heteroscedasticity. From the model, the exhibited MIC of ceftazidime was significantly higher in Chile, Croatia, the Dominican Republic, Greece, Guatemala, India, Italy, South Korea, Latvia, Nigeria, Panama, Philippines, Poland, Romania, Russia, Taiwan, Thailand, and Ukraine. Contrarily, it was significantly lower in Australia, Belgium, Columbia, Czech Republic, Denmark, France, Germany, Ireland, Japan, The Netherlands, and Spain compared to Argentina. In the case of bacterial strain, MIC of ceftazidime showed significant variations. Specifically, in *Enterobacter* spp. and *Providencia* spp., ceftazidime’s MIC was significantly higher by 2.33 μg/mL and 1.44 μg/mL, respectively. However, in *Klebsiella* spp. and *Proteus* spp., it was significantly lower by 0.73 μg/mL and 0.28 μg/mL, respectively, compared to *E. coli*. The MIC of ceftazidime was found to be significantly higher in the Medicine ICU and Pediatric ICU compared to the clinic or office settings. Additionally, for every one-year increase, the MIC of ceftazidime decreased by 0.97 μg/mL. In the presence of the *bla*_CTX-M-1_ and *bla*_CTX-M-2_ groups, the MIC of ceftazidime was significantly increased by 2.97 μg/mL and 1.54 μg/mL, respectively. Conversely, in the presence of *bla*_CTX-M-8/25_ and *bla*_CTX-M-9_, it decreased significantly by 0.34 μg/mL and 0.58 μg/mL, respectively. Similarly, the presence of *bla*_VEB_, *bla*_PER_, *bla*_GES_, *bla*_ACC_, *bla*_CMY-2_, *bla*_DHA_, and *bla*_FOX_ genes significantly increased the MIC of ceftazidime by 6.23 μg/mL, 7.44 μg/mL, 2.17 μg/mL, 9.62 μg/mL, 3.29 μg/mL, 2.14 μg/mL, and 2.12 μg/mL, respectively. Furthermore, ceftazidime’s MIC showed a substantial increase of 3.44 μg/mL, 1.23 μg/mL, 2.57 μg/mL, 5.77 μg/mL, 3.51 μg/mL, and 2.46 μg/mL in the presence of *bla*_KPC_, *bla*_OXA_, *bla*_NDM_, *bla*_IMP_, *bla*_VIM_, and *bla*_SHV_ genes, respectively (Table 1).

#### 2.1.3. Ceftaroline

Robust model was used in the ceftaroline model due to heteroscedasticity. From the model, the MIC of ceftaroline demonstrated significantly lower in Brazil, Cameroon, Canada, Chile, China, Colombia, Costa Rica, Croatia, the Dominican Republic, Finland, France, Germany, Guatemala, Hong Kong, India, Ireland, Ivory Coast, Japan, Jordan, Kenya, Latvia, Lithuania, Malaysia, Morocco, the Netherlands, New Zealand, Nigeria, Panama, Portugal, Qatar, Romania, Saudi Arabia, Singapore, Spain, Sweden, Switzerland, Ukraine, the United Kingdom, and the United States compared to Argentina. Regarding bacterial strains, ceftaroline MIC was significantly higher in *Citrobacter* spp., *Enterobacter* spp., *Klebsiella* spp., *Morganella* spp., and *Providencia* spp. compared to *E. coli*. The MIC of ceftaroline was found to be higher in the Medicine General and ICU units compared to clinic or office settings. Additionally, with every one-year increase, the MIC of ceftaroline decreased by 0.65 μg/mL. In the presence of the *bla*_CTX-M-1_, *bla*_CTX-M-2_, *bla*_CTX-M-8/25_, and *bla*_CTX-M-9_ gene groups, the MIC increased by 3.09 μg/mL, 2.70 μg/mL, 3.07 μg/mL, and 3.07 μg/mL, respectively. Furthermore, in the presence of carbapenemases genes (*bla*_KPC_, *bla*_NDM_, *bla*_IMP_, and *bla*_VIM_), the MIC of ceftaroline increased by 1.98 μg/mL, 1.06 μg/mL, 2.37 μg/mL, and 1.58 μg/mL, respectively (Table 1).

#### 2.1.4. Imipenem

Robust model was used in the imipenem model due to heteroscedasticity. From the model, the MIC of imipenem was significantly higher in Australia, Austria, Belgium, Brazil, Canada, Chile, China, Croatia, Denmark, Germany, Greece, Guatemala, Hong Kong, Hungary, India, Ireland, Italy, Kuwait, Latvia, Lithuania, Malaysia, Mexico, Morocco, The Netherlands, Nigeria, Qatar, Romania, Russia, Saudi Arabia, Spain, Sweden, Switzerland, Taiwan, Thailand, Turkey, Ukraine, and the United States compared to Argentina. The MIC of imipenem exhibited significant variations across different bacterial strains. Specifically, in *Citrobacter* spp., *Enterobacter* spp., *Klebsiella* spp., *Morganella* spp., *Proteus* spp., *Providencia* spp., *Raoultella* spp., and *Serratia* spp., the MIC was significantly higher by 2.04 μg/mL, 3.03 μg/mL, 1.63 μg/mL, 2.30 μg/mL, 9.98 μg/mL, 3.55 μg/mL, 2.35 μg/mL, and 4.67 μg/mL, respectively, compared to *E. coli*. When considering age groups, individuals between 19 and 64 years and those aged 85 years and over had significantly higher MIC values compared to the 0 to 2 years age group. Furthermore, the MIC of imipenem was found to be significantly higher in the Medicine General, Medicine ICU, Surgery General, and Surgery ICU compared to clinic or office settings. Moreover, with every one-year increase, the MIC of imipenem increased by 1.03 μg/mL. In the presence of the *bla*_CTX-M-1_ and *bla*_CTX-M-9_ gene groups, the MIC of imipenem decreased by 0.81 μg/mL and 0.84 μg/mL, respectively. Conversely, the presence of *bla*_GES_, *bla*_CMY-2_, and *bla*_DHA_ genes increased the MIC by 1.61 μg/mL, 1.46 μg/mL, and 2.20 μg/mL, respectively. Additionally, in the presence of carbapenemase genes (*bla*_KPC_, *bla*_OXA_, *bla*_NDM_, *bla*_IMP_, and *bla*_VIM_), the MIC of imipenem significantly increased by 17.1 μg/mL, 7.46 μg/mL, 10.1 μg/mL, 5.39 μg/mL, and 7.76 μg/mL, respectively (Table 1).

#### 2.1.5. Meropenem

A robust model was used in the meropenem model due to heteroscedasticity. From the model, the MIC of meropenem was significantly higher in Australia, Belgium, Brazil, Canada, Chile, China, Costa Rica, Croatia, Greece, Guatemala, India, Italy, Kenya, South Korea, Kuwait, Malaysia, Mexico, Nigeria, the Philippines, Poland, Romania, Russia, Saudi Arabia, South Africa, Spain, Taiwan, Thailand, Turkey, Ukraine, the United States, and Venezuela compared to Argentina. In the context of bacterial strains, MIC was found to be significantly higher in *Citrobacter* spp., *Enterobacter* spp., *Klebsiella* spp., *Proteus* spp., *Providencia* spp., and *Serratia* spp. compared to *E. coli*. In terms of age groups, the MIC of meropenem was higher in the 19 to 64 Years, 65 to 84 Years, and 85 and over age groups. The Medicine and Surgery ICU settings exhibited higher MIC values for meropenem compared to clinic or office settings. Furthermore, with each one-year increase in age, the MIC of meropenem increased by 1.09 μg/mL. In the presence of *bla*_CTX-M-1_ and *bla*_CTX-M-9_ gene groups, the MIC of meropenem decreased by 0.87 μg/mL and 0.83 μg/mL, respectively. On the other hand, in the presence of carbapenemase genes (*bla*_KPC_, *bla*_OXA_, *bla*_NDM_, *bla*_IMP_, and *bla*_VIM_), the MIC of meropenem increased significantly by 67.38 μg/mL, 21.21 μg/mL, 37.68 μg/mL, 19.40 μg/mL, and 15.05 μg/mL, respectively (Table 1).

### 2.2. Prediction of MIC: Retail Meat Dataset from NARMS

#### 2.2.1. Ceftriaxone

The heteroscedasticity was detected in the initial model of ceftriaxone dataset as such, weighted least square model was used in the predictive modeling. The MIC of ceftriaxone was significantly higher in the presence of *bla*_CMY-2_ by 57.59 μg/mL. In addition, in the presence of ESBL genes, *bla*_CTX-M-1_, *bla*_CTX-M-55_, and *bla*_CTX-M-65_; MIC of ceftriaxone significantly increased by 223.23 μg/mL, 254.64 μg/mL, and 203.68 μg/mL respectively. There was no significant difference in MIC in the following years compared to 2002, except in 2009 and 2010 where MIC increased by 1.10 μg/mL and 1.49 μg/mL, respectively (Figure 1) (Appendix A).

#### 2.2.2. Cefoxitin

Weighted least squares model was used in the cefoxitin model due to heteroscedasticity. The MIC of cefoxitin increased significantly in the presence of *bla*_CTX-M-65_ and *bla*_TEM-1_ by 1.57 μg/mL and 1.04 μg/mL respectively. Compared to *E. coli* isolates, the MIC of cefoxitin in *Salmonella enterica* isolates decreased significantly by 0.67 μg/mL (Figure 2). Interaction was observed between *bla*_CMY-2_ and year; *bla*_CMY-2_ and meat type; *bla*_CMY-2_ and genus; genus, and meat type for Cefoxitin in retail meat data. Over the 17 years, the MIC of cefoxitin increased by an average of 8.66 μg/mL in the presence of the *bla*_CMY-2_ gene (Figure 3). Moreover, MIC increased in *E. coli* and *S. enterica* by 5.05 μg/mL and 6.76 μg/mL respectively in the presence of *bla*_CMY-2_ (Figure 4). In the presence of the *bla*_CMY-2_ gene, the MIC value increased in chicken breast, ground beef, ground turkey, and pork cut at the level of 5.04 μg/mL, 4.36 μg/mL, 4.59 μg/mL, and 3.77 μg/mL, respectively (Figure 5). Finally, in the presence of *E. coli*, the MIC of cefoxitin significantly increased in chicken breast and ground turkey by 1.49 μg/mL and 1.31 μg/mL respectively (Figure 6) (Appendix A).

#### 2.2.3. Ceftiofur

A weighted least squares model was used in the ceftiofur model due to heteroscedas-ticity. The MIC of ceftiofur increased significantly in the presence of *bla*_CTX-M-1_, *bla*_CTX-M-65_, *bla*_SHV-2_, and *bla*_TEM-1_ by 8.82 μg/mL, 9.11 μg/mL, 8.18 μg/mL, and 1.04 μg/mL respectively. In comparison with *E. coli*, the MIC of ceftiofur significantly increased in *S. enterica* by 2.28 μg/mL (Figure 7). There was an interaction found between *bla*_CMY-2_ and year; and *bla*_CMY-2_, and meat type. The average increase of the MIC of ceftiofur by 10.20 μg/mL in the presence of the *bla*_CMY-2_ gene over 14 years from 2002 to 2015 (Figure 8). In addition, In the presence of the *bla*_CMY-2_ gene, the MIC of ceftiofur value increased in chicken breast, ground beef, ground turkey, and pork cuts by 14.46 μg/mL, 12.49 μg/mL, 12.89 μg/mL, and 9.63 μg/mL respectively (Figure 9) (Appendix A).

## 3. Discussion

This study focuses on using predictive modeling to estimate the phenotypic MIC values of beta-lactam antimicrobials to members of the family Enterobacteriaceae from different beta-lactamase resistance genes. We utilized the enhanced human and retail meat data set provided by the ATLAS and NARMS surveillance programs, respectively. To achieve accurate predictive modeling, we used a robust/weighted least square linear regression framework that was described earlier in research [28,29,30,31], which has proven successful in predicting and validating MICs based on the presence of beta-lactamase genes while adjusting for other epidemiological variables. 

Antimicrobial treatment decisions typically hinge on culture and susceptibility testing; disc diffusion method or minimum inhibitory concentration [32]. However, by employing a predictive approach and leveraging insights about the prevalent bacteria in a specific geographic region, the results derived from predictive models can inform initial treatment choices while awaiting the outcomes of culture and susceptibility testing. This predictive modeling approach is highly advantageous and holds significant potential, complementing traditional laboratory testing in both diagnosis and treatment strategies. By taking into consideration the practice-based AMR trend, the predictive approach provides evidence-based prioritization of antimicrobial selection even with the timely availability of culture and susceptibility results. In this study, within the human dataset, the MIC of cephalosporins and carbapenems antimicrobials exhibited an annual increase ranging from 0.65 to 1.09 μg/mL. This trend could be attributed to the yearly escalation in antimicrobial usage. Klein et al. reported a 65% increase in antibiotic consumption or defined daily doses and a 39% rise in antibiotic consumption rates between 2000 and 2015 [33]. Geographical location plays a significant role in influencing the rise in antimicrobial susceptibility. In our study, we observed an increase in MIC primarily driven by low and middle-income countries. This finding aligns with previous research that indicates resistance levels are exacerbated by the widespread and inappropriate use of antibiotics in humans, animals, and crops as well as the inadequate management of pharmaceutical waste [34]. However, in this research, the predicted MIC value for gender was not found to be statistically significant. This result contrasts with the previous findings of Schröder et al., who reported that women had a 27% higher likelihood of receiving antibiotic prescriptions during their lifetime and exhibited greater antimicrobial resistance [35]. Furthermore, our investigation revealed that medicine and surgery intensive care units (ICUs) displayed elevated resistance levels to cephalosporin and carbapenem drugs. This phenomenon can be attributed to a multitude of factors, including a prolonged stay, escalated antibiotic consumption, frequent exposure to healthcare-associated infections, the potential for antibiotics to be excessively or improperly administered, and the rapid dissemination of multidrug resistance genes and mutations [36]. 

One major challenge for the clinical settings is that the definition of susceptibility and resistance breakpoint for susceptibility testing is continuously evolving or in some cases non-available. Also, the phenotypic method conventionally requires breakpoint value and interpretation for therapeutic purposes. However, predictive modeling as performed in this study provides a framework for clinical and therapeutic decision-making based on antimicrobial sensitivity testing (AST) or genomic data integration without the need for clinical breakpoint. The presence of beta-lactamase genes in bacteria of the family Enterobacteriaceae provides a resistance mechanism that inactivates the beta-lactam antimicrobials [37]. These beta-lactamase inactivations directly translate into increases in MIC values, which can indirectly provide evidence of the extent of antimicrobial resistance. For example, and as shown in this study, in the presence of carbapenemase (*bla*_KPC_, *bla*_OXA_, *bla*_NDM_, *bla*_IMP_, *bla*_VIM_), the MIC of meropenem increased by values from 15.5–67.38 μg/mL even though the laboratory MIC testing range of meropenem is 0.06–4 μg/mL. Similarly, Imipenem increased by 5.39–17.1 μg/mL in the presence of carbapenemase genes, where ≥4 μg/mL is considered as the resistance for Enterobacteriaceae [38]. Such increases are evidently outside the resistance range and directly inferred resistance even without the need for a breakpoint.

From the NARMS retail meat dataset, we have seen significant and large increases in the range of 57.59–254.64 μg/mL in the MICs of ceftriaxone in *E. coli* and *S. enterica* in the presence of extended-spectrum beta-lactamase genes; *bla*_CMY-2_, *bla*_CTX-M-1_, *bla*_CTX-M-55_, and *bla*_CTX-M-65_ even though the laboratory MIC testing range of ceftriaxone is 0.25–64 μg/mL. The primary mechanism of ceftriaxone resistance in *E. coli* and *Salmonella enterica* is the production of ESBL genes, and ceftriaxone resistance is one of the indicator of the presence of ESBL production [39]. On the other hand, the presence of the ampC beta-lactamase gene (*bla*_CMY-2_) exhibited resistance and increased MIC of cefoxitin by an average of 8.66 μg/mL from 2002 to 2017 in the United States. This result can be correlated to the transfer of *bla*_CMY-2_ genes in food animals over the years. It has been proved that the *bla*_CMY-2_ plasmid can be transferred between *E. coli* and *S. enterica* isolates originating from both food animals and humans [40]. In our study, the presence of the *bla*_CMY-2_ gene was observed to significantly elevate resistance levels to cefoxitin and ceftiofur in retail meat, with MIC increasing by 3.77–5.04 μg/mL and 9.63–14.46 μg/mL, respectively. Previously, the *bla*_CMY-2_ gene had been identified in *E. coli* and *S. enterica* isolates obtained from food-producing animals and retail meat, displaying complete resistance to cefoxitin and an 88% resistance rate to ceftiofur [41]. 

With the availability of historical phenotypic MIC and genotypic data, we can accurately predict and provide a more accurate expected antimicrobial in vitro and in vivo response clinically. Integrating phenotypic and genotypic data not only provides optimal clinical and therapeutic decision-making not based on AST or genomic method alone but also avoids errors associated with the clinical interpretations of AST or genomic method alone [42]. Other impacts of this work include the provision of opportunity for monitoring AMR trends taking into consideration other factors for both human and food animal/retail meat that may influence AMR trends, e.g., across hospital specialties, bacteria type, gender, and age group as well as across time and geographical space. Since the increase in MIC indirectly translates into the possibility of resistance development, the increase in MIC trend over time and geographical space as shown in this predictive modeling can be used for AMR monitoring. As recommended by the WHO, AMR monitoring is germane for supporting antimicrobial stewardship policy-making at national and global scales [11]. Also, such monitoring provides the framework for a better understanding of the epidemiology for the prevention, and control of antimicrobial resistance to critically important antimicrobials not only in human health but across one health interface.

In addition, the ability to predict beta-lactam phenotypic antimicrobial susceptibility directly from beta-lactamase resistance genes further illuminates how these resistance genes mediate resistance to critically important beta-lactam antimicrobials. Such predictive modeling may help reduce the reliance on routine phenotypic testing with higher turnaround times in diagnostic, therapeutic, and surveillance of antimicrobial-resistant bacteria of the family Enterobacteriaceae.

This study has certain limitations, as the data analysis was conducted based on the availability of data. We incorporated left and right-censored MIC values into the true MIC values, with left-censored values representing the smallest concentration and right-censored MIC values representing the highest concentration. This study was only focused on resistant genes of beta-lactam antimicrobials in the Enterobacteriaceae bacteria family and application of these models to other bacteria families may be impractical.

## 4. Materials and Methods

### 4.1. Data Collection

A cross-sectional study design was performed, and the data for this research were retrieved from two AMR surveillance programs; ATLAS (access at https://www.pfizer.com/science/focus-areas/anti-infectives/antimicrobial-surveillance, accessed on 15 June 2023) and obtained through the Vivli Center for Global Clinical Research Data (accessed at https://amr.vivli.org on 15 June 2023), and the retail meat surveillance data from the NARMS (accessed at https://www.cdc.gov/narms/index.html on 1 February 2022). 

Data from the ATLAS program was collected from 61 countries and involved samples obtained from humans across various hospital specialties and six different age categories: 0–2 years, 3–12 years, 13–18 years, 19–64 years, 65–84 years, and ≥85 years. On the other hand, in NARMS surveillance, USDA collected retail meat samples of cattle, chicken, swine, and turkey in the United States. 

Data for the period 2004 to 2021 was acquired from ATLAS. The available data included bacterial species of the family Enterobacteriaceae (*E. coli*, *Citrobacter* spp., *Enterobacter* spp., *Klebsiella* spp., *Morganella* spp., *Proteus* spp., *Providencia* spp., *Raoultella* spp., and *Serratia* spp.), various beta-lactamase gene groups (*bla*_CTXM-1_, *bla*_CTXM-2_, *bla*_CTXM-8/25_, *bla*_CTXM-9_), and specific genes *bla*_SHV_, *bla*_TEM_, *bla*_VEB_, *bla*_PER_, *bla*_GES_, *bla*_ACC_, *bla*_CMY-2_, *bla*_DHA_, *bla*_FOX_, *bla*_ACT_, *bla*_KPC_, *bla*_OXA_, *bla*_NDM_, *bla*_IMP_, and *bla*_VIM_). Additionally, we collected information on the minimum inhibitory concentration (MIC) of cefepime (fourth-generation cephalosporin), ceftazidime (third-generation cephalosporin), ceftaroline (fifth-generation cephalosporin), imipenem, and meropenem antimicrobial drugs. In the case of ATLAS, MICs were established through broth microdilution in accordance with Clinical and Laboratory Standards Institute (CLSI) guidelines, and the interpretation utilized the 2020 CLSI breakpoints. For NARMS, MIC breakpoints followed the guidelines established by the CDC NARMS [38], available at (https://www.cdc.gov/narms/antibiotics-tested.html, accessed on 1 February 2022). From the NARMS data, observations of two bacteria (*Salmonella enterica* and *E. coli*) and four types of meat (chicken breast, ground beef, ground turkey, and pork cut) were obtained for the period 2002 to 2018. To predict the MICs of three drugs (ceftriaxone, cefoxitin, and ceftiofur), several beta-lactamase genes were also acquired from the dataset. These genes included (*bla*_CMY-2_, *bla*_CMY-3_, *bla*_CTX-M-1_, *bla*_CTX-M-55_, *bla*_CTX-M-65_, *bla*_SHV-2_, *bla*_TEM-1_).

### 4.2. Data Analysis

We used Microsoft Excel 365 for data cleaning, organization, and management. From the raw data source, the beta-lactamase genes were extracted and assembled with other variables. Independent variables in the human dataset from ATLAS included country, bacterial species, gender, age group, specialty (the unit where patients were treated), year, and beta-lactamase genes. On the other hand, the dependent variables were the MIC values of cefepime, ceftazidime, ceftaroline, imipenem, and meropenem. For the NARMS dataset, bacterial species, year, meat types, and beta-lactamase genes were the independent variables, and the MIC values of ceftriaxone, cefoxitin, and ceftiofur were the dependent variables. The focus of our analysis was to predict MICs of Enterobacteriaceae from different beta-lactamase resistance gene families available in the ATLAS and NARMS data while adjusting for the other associated variables. 

To conduct our statistical analysis, the dataset was exported into the R open-source scripting software (version 4.2.2, R Foundation for Statistical Computing, Vienna, Austria https://www.R-project.org/, accessed on 14 February 2022). Since the MIC values of the antimicrobials (cefepime, ceftazidime, ceftaroline, ceftriaxone, cefoxitin, ceftiofur, imipenem, and meropenem) followed a geometric increase pattern, we transformed the MICs using log2 transformation to approximate the MIC values to a normal distribution as much as possible. Initially, simple linear regression modeling was performed to explore the relationship between the log2-MIC of each antimicrobial and the other variables individually. Subsequently, a multivariable linear regression model was employed with each antimicrobial as the dependent variable. Multicollinearity and interaction between the variables were checked. In addition, all models were checked based on the higher adjusted R square and lower Akaike information criterion (AIC) for selecting the final model. If heteroscedasticity is detected in the initial model, indicating that the variability of the errors is not constant across all levels of the independent variables, it will leading to a transition to a weighted least squares model (WLS) [43]. It assigned different weights to different observations based on the variance of the errors. The model lowered the error variability by giving lower weights to observations with more variability and higher weights to lower variance. WLS model followed the equation:$$\sum_{i=1}^{n}{w_{i}\;(y_{i}-{\hat{y}}_{i})}^{2}$$
where *n* is the number of observations, $y_{i}$ is the observed response for the *i*-th observation, ${\hat{y}}_{i}$ is the predicted response for the *i*-th observation and $w_{i}$ is the weight assigned to the *i*-th observation. To determine the weights $\left(w_{i}\right)$ the inverse of the squared residuals ($\epsilon_{i})\;$was applied.
$$w_{i}=\frac{1}{{\epsilon_{i}}^{2}}$$

The retail meat dataset conformed to the weighted least squares model, whereas the human dataset did not, necessitating a switch to a robust model. The coefficients of the weighted least squares or robust models were back-transformed by computing the square of log2-MIC values. ATLAS data were presented in a table, and NARMS data were visualized in plots using R package sjPlot. For all statistical analyses, statistical significance was set at *p* < 0.05.

## 5. Conclusions

The challenge of identifying and diagnosing antimicrobial resistance (AMR) is a significant concern for global health. Timely and precise diagnosis is a crucial factor in the battle against AMR, with one of the primary obstacles being the lengthy process involved in determining susceptibility levels. The current gold standard for susceptibility testing, known as the MIC method, is both time-consuming, labor-intensive, and expensive. However, the application of predictive modeling, specifically using multivariable linear regression analysis, has shown promising results in validating anticipated MIC values based on factors such as beta-lactamase genes and other epidemiological variables. Unlike conventional breakpoint interpretation of phenotypic values, this predictive modeling offers a more accurate computed value, particularly in the presence of specific beta-lactamase genes in the integrated surveillance dataset. Utilizing this type of analysis enables us to identify the most suitable antimicrobial treatments and employ them in combination therapies, ultimately leading to improved outcomes for patients facing antimicrobial resistance.

## Figures and Tables

**Figure 1 antibiotics-13-00224-f001:**
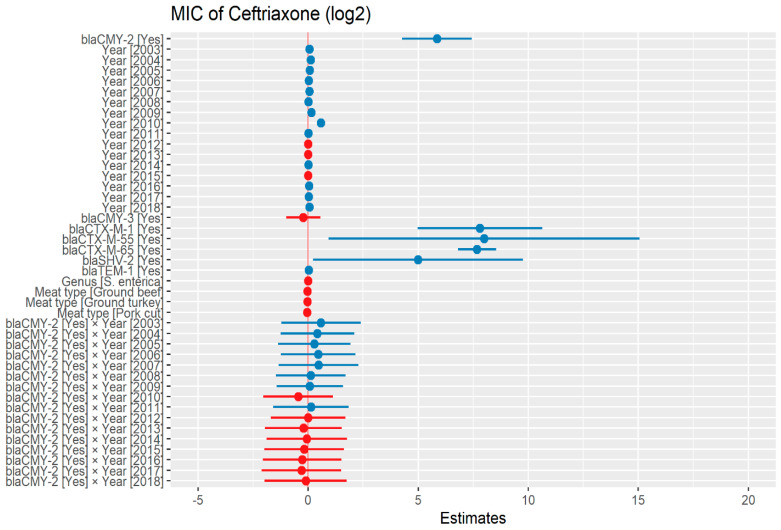
Plot showing prediction of MIC of ceftriaxone using the weighted least squares (WLS) model in retail meat dataset from NARMS surveillance. The estimates are represented in log2 values. The interaction between two variables is represented by (×). The red color indicated the decreased MIC values, while the blue color represented the increased MIC values.

**Figure 2 antibiotics-13-00224-f002:**
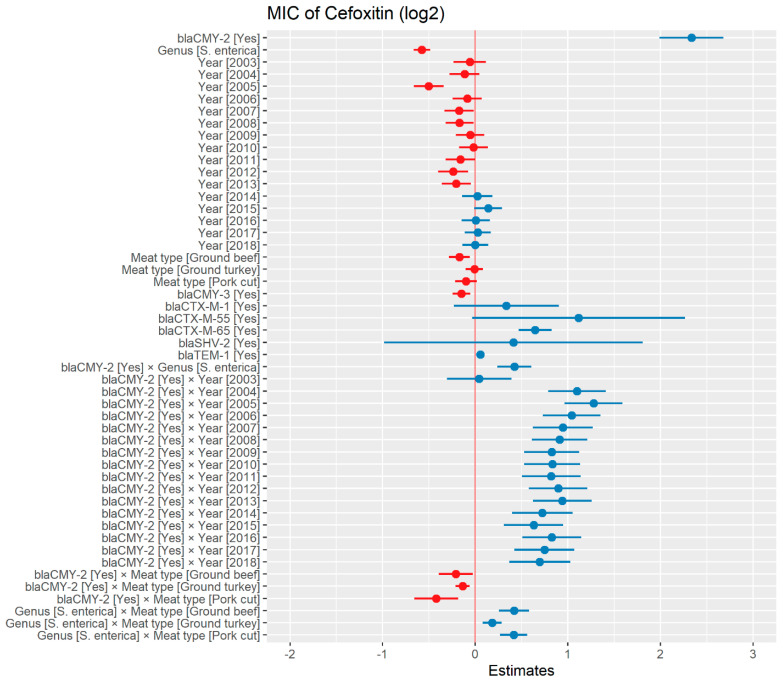
Plot showing prediction of MIC of cefoxitin using the weighted least squares (WLS) model in retail meat dataset from NARMS surveillance. The estimates are represented in log2 values. The interaction between two variables is represented by (×). The red color indicated the decreased MIC values, while the blue color represented the increased MIC values.

**Figure 3 antibiotics-13-00224-f003:**
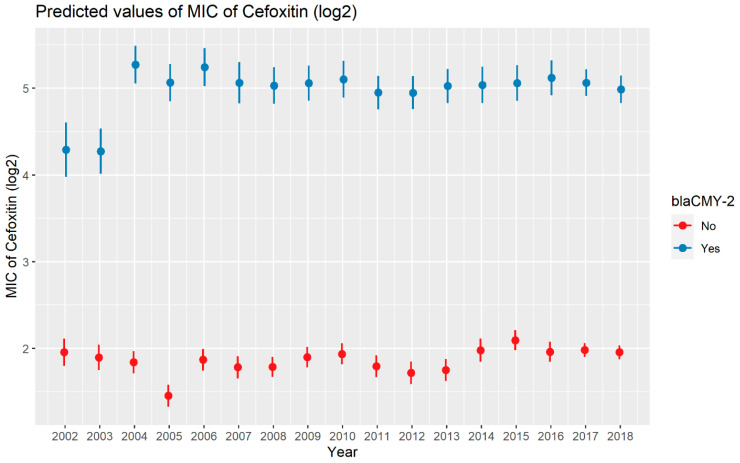
Plot showing prediction of MIC of cefoxitin with the interaction of *bla*_CMY-2_ and years. The estimates are represented in log2 values.

**Figure 4 antibiotics-13-00224-f004:**
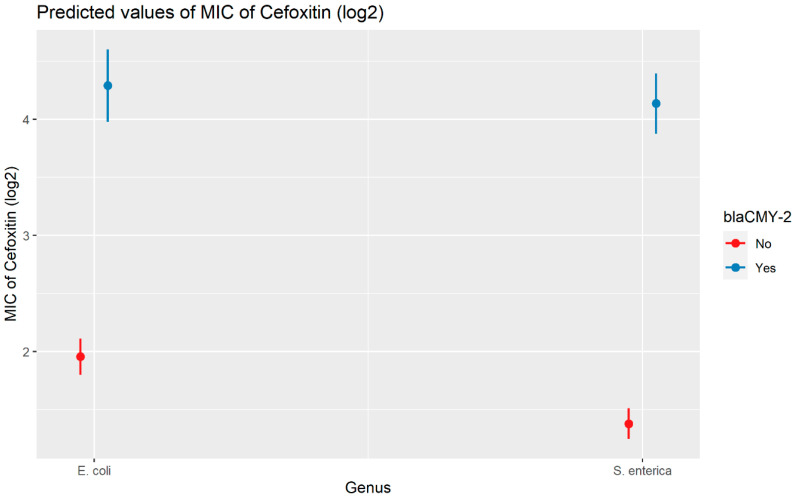
Plot showing prediction of MIC of cefoxitin with the interaction of *bla*_CMY-2_ and bacterial genus. The estimates are represented in log2 values.

**Figure 5 antibiotics-13-00224-f005:**
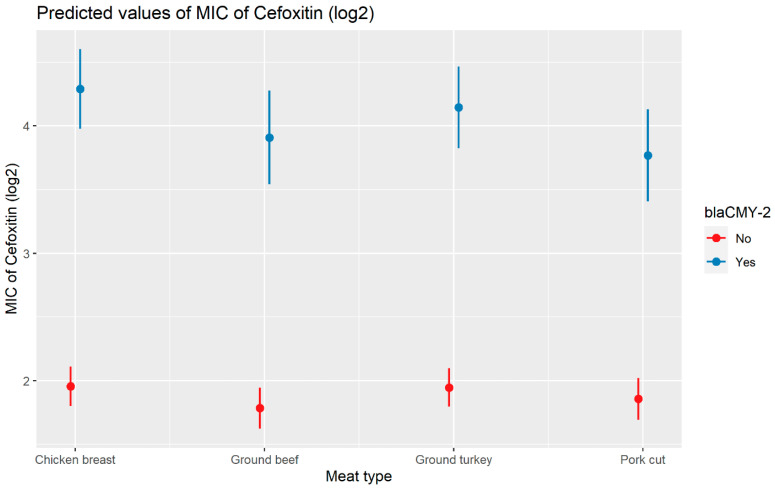
Plot showing prediction of MIC of cefoxitin with the interaction of *bla*_CMY-2_ and meat types. The estimates are represented in log2 values.

**Figure 6 antibiotics-13-00224-f006:**
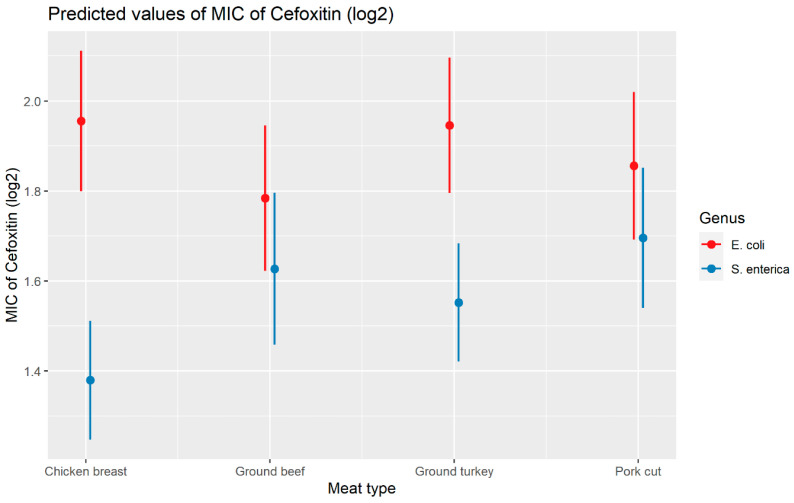
Plot showing prediction of MIC of cefoxitin with the interaction of bacterial genus and meat types. The estimates are represented in log2 values.

**Figure 7 antibiotics-13-00224-f007:**
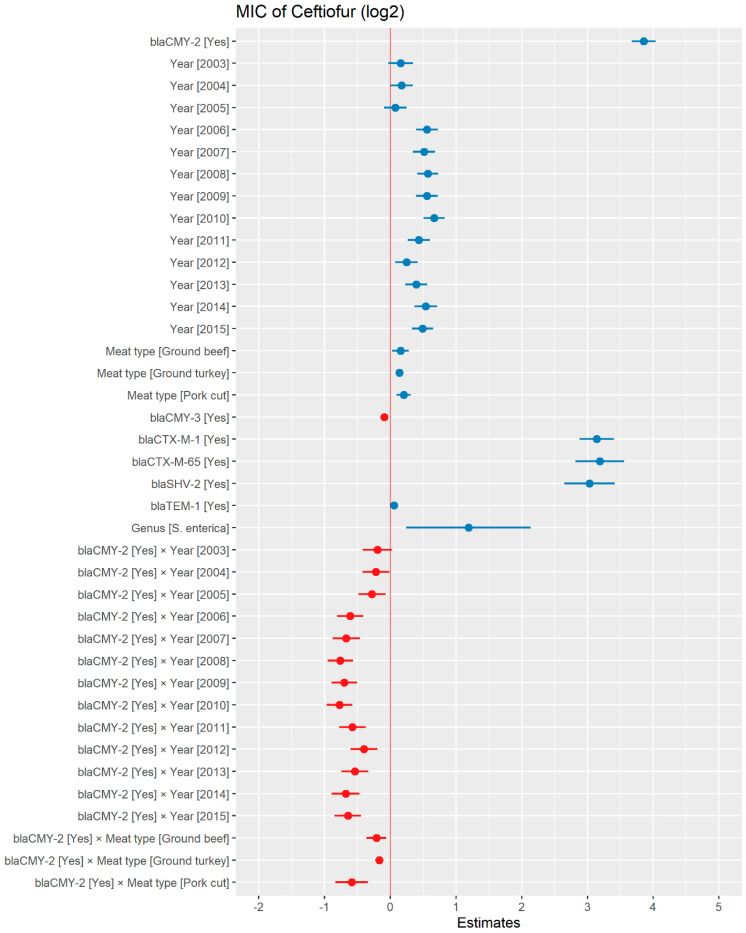
Plot showing prediction of MIC of ceftiofur using the weighted least squares (WLS) model in retail meat dataset from NARMS surveillance. The estimates are represented in log2 values. The interaction between two variables is represented by (×). The red color indicated the decreased MIC values, while the blue color represented the increased MIC values.

**Figure 8 antibiotics-13-00224-f008:**
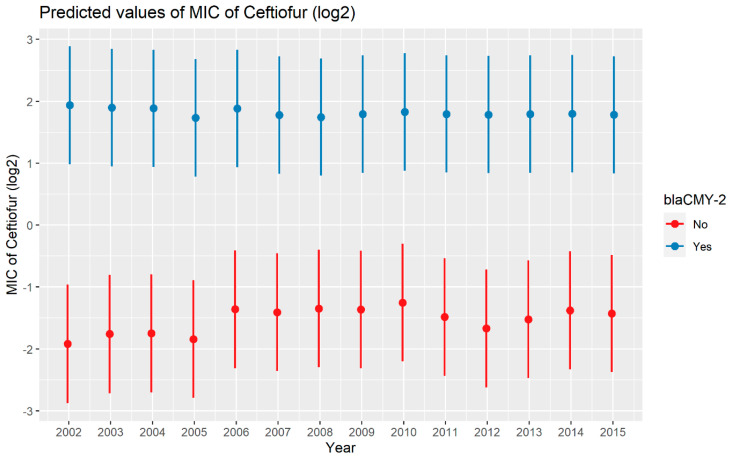
Plot showing prediction of MIC of ceftiofur with the interaction of *bla*_CMY-2_ and years. The estimates are represented in log2 values.

**Figure 9 antibiotics-13-00224-f009:**
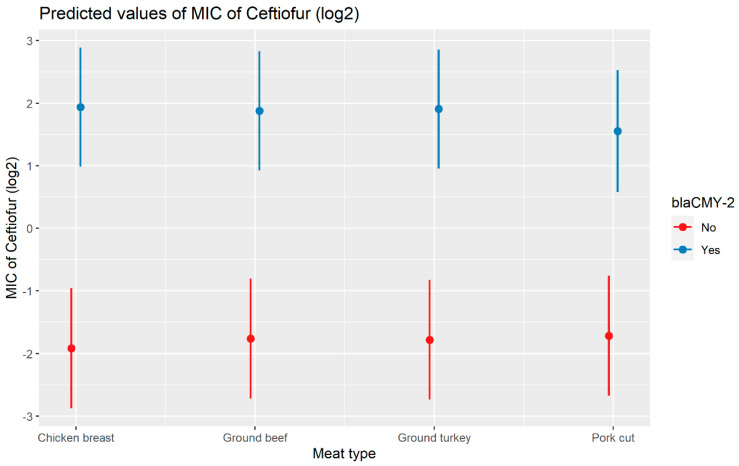
Plot showing prediction of MIC of ceftiofur with the interaction of *bla*_CMY-2_ and meat types. The estimates are represented in log2 values.

**Table 1 antibiotics-13-00224-t001:** Prediction of MIC of selected antimicrobials in the human dataset from ATLAS surveillance.

Variables	Categories	Cefepime			Ceftazidime			Ceftaroline			Imipenem			Meropenem		
		MIC	95% CI	*p* Value	MIC	95% CI	*p* Value	MIC	95% CI	*p* Value	MIC	95% CI	*p* Value	MIC	95% CI	*p* Value
Country	Argentina	Ref														
	Australia	0.75	0.65–0.99	<0.01	0.64	0.52–0.77	<0.01	0.81	0.66–0.98	0.03	1.12	1.01–1.23	0.02	1.15	1.01–1.31	0.03
	Austria	0.80	0.77–0.96	0.04	0.72	0.54–0.95	0.02	0.71	0.53–0.94	0.02	1.17	1.01–1.34	0.03	1.13	0.95–1.34	0.14
	Belgium	0.86	0.87–1.04	0.01	0.80	0.70–0.92	<0.01	1.00	0.88–1.14	0.94	1.10	1.01–1.19	0.02	1.18	1.05–1.32	<0.01
	Brazil	0.95	0.69–1.38	0.26	0.95	0.86–1.05	0.36	0.87	0.79–0.96	0.01	1.17	1.09–1.25	<0.01	1.35	1.22–1.49	<0.01
	Cameroon	0.98	0.67–0.95	0.90	0.89	0.68–1.16	0.42	0.61	0.52–0.70	<0.01	1.11	0.92–1.35	0.28	0.80	0.66–0.97	0.02
	Canada	0.80	0.96–1.15	0.01	0.83	0.68–1.01	0.08	0.57	0.49–0.65	<0.01	1.17	1.05–1.31	0.01	1.26	1.09–1.46	<0.01
	Chile	1.06	1.03–1.32	0.24	1.16	1.02–1.31	0.02	0.86	0.77–0.96	0.01	1.22	1.13–1.31	<0.01	1.84	1.65–2.06	<0.01
	China	1.17	0.74–0.89	0.01	0.96	0.81–1.14	0.65	0.50	0.44–0.56	<0.01	1.37	1.24–1.51	<0.01	1.33	1.18–1.49	<0.01
	Colombia	0.82	0.74–1.47	<0.01	0.73	0.65–0.82	<0.01	0.90	0.81–0.99	0.04	1.02	0.95–1.09	0.57	0.98	0.89–1.09	0.75
	Costa Rica	1.04	1.05–1.38	0.82	0.85	0.55–1.32	0.48	0.63	0.54–0.74	<0.01	1.06	0.88–1.29	0.53	1.34	1.03–1.74	0.03
	Croatia	1.20	0.87–1.06	0.01	1.42	1.21–1.68	<0.01	0.62	0.56–0.69	<0.01	1.34	1.20–1.48	<0.01	1.44	1.24–1.68	<0.01
	Czech Republic	0.96	0.46–0.70	0.42	0.82	0.73–0.93	<0.01	1.04	0.93–1.17	0.47	1.04	0.97–1.12	0.29	0.89	0.81–0.98	0.02
	Denmark	0.57	0.83–1.28	<0.01	0.36	0.26–0.49	<0.01	0.77	0.55–1.07	0.12	1.17	1.01–1.34	0.03	0.95	0.81–1.09	0.46
	Dominican Republic	1.03	0.32–1.86	0.79	1.42	1.16–1.72	<0.01	0.53	0.46–0.60	<0.01	0.93	0.82–1.06	0.27	1.03	0.93–1.15	0.59
	Finland	0.77	0.71–0.87	0.56	0.82	0.35–1.93	0.66	0.49	0.25–0.97	0.04	0.99	0.74–1.34	0.96	1.04	0.85–1.28	0.70
	France	0.78	0.75–0.92	<0.01	0.75	0.66–0.85	<0.01	0.87	0.77–0.98	0.02	1.06	0.99–1.14	0.11	0.99	0.91–1.09	0.87
	Germany	0.83	1.16–1.41	<0.01	0.65	0.56–0.74	<0.01	0.86	0.76–0.97	0.01	1.16	1.07–1.25	<0.01	1.09	0.98–1.21	0.11
	Greece	1.28	0.96–1.25	<0.01	1.25	1.10–1.40	<0.01	1.08	0.97–1.21	0.17	1.27	1.18–1.37	<0.01	1.69	1.53–1.87	<0.01
	Guatemala	1.09	0.53–0.93	0.19	1.24	1.06–1.44	0.01	0.60	0.54–0.67	<0.01	1.37	1.24–1.52	<0.01	1.76	1.53–2.02	<0.01
	Hong Kong	0.70	0.85–1.03	0.01	0.79	0.58–1.06	0.12	0.62	0.49–0.79	<0.01	1.35	1.13–1.61	<0.01	1.20	0.99–1.46	0.06
	Hungary	0.94	1.16–1.37	0.20	1.01	0.88–1.14	0.92	1.00	0.89–1.13	0.99	1.12	1.04–1.22	0.01	1.07	0.97–1.19	0.18
	India	1.26	0.50–0.88	<0.01	1.38	1.25–1.53	<0.01	0.64	0.59–0.69	<0.01	1.30	1.21–1.41	<0.01	1.79	1.61–1.99	<0.01
	Ireland	0.67	0.89–1.08	<0.01	0.69	0.52–0.91	0.01	0.54	0.44–0.67	<0.01	1.17	1.00–1.36	0.05	1.07	0.88–1.30	0.48
	Israel	0.98	1.09–1.28	0.72	0.97	0.86–1.08	0.62	1.07	0.96–1.19	0.20	1.04	0.97–1.11	0.29	1.03	0.94–1.13	0.51
	Italy	1.18	0.72–1.20	<0.01	1.44	1.30–1.58	<0.01	1.02	0.93–1.12	0.62	1.19	1.12–1.27	<0.01	1.45	1.33–1.58	<0.01
	Ivory Coast	0.93	0.67–0.95	0.59	0.96	0.74–1.24	0.75	0.68	0.59–0.79	<0.01	0.97	0.83–1.14	0.71	0.86	0.74–0.99	0.04
	Japan	0.80	0.55–1.10	0.01	0.67	0.54–0.82	<0.01	0.72	0.58–0.89	<0.01	1.11	0.99–1.23	0.06	1.11	0.99–1.25	0.08
	Jordan	0.78	1.01–1.35	0.16	1.15	0.88–1.48	0.30	0.44	0.38–0.52	<0.01	0.93	0.73–1.19	0.59	0.79	0.58–1.08	0.14
	Kenya	1.17	0.99–1.21	0.04	1.23	0.98–1.55	0.07	0.81	0.68–0.95	0.01	0.87	0.69–1.09	0.24	1.36	1.11–1.67	<0.01
	South Korea	1.10	0.87–1.04	0.06	1.19	1.06–1.34	<0.01	1.01	0.90–1.12	0.91	1.10	1.03–1.18	0.01	1.19	1.09–1.31	<0.01
	Kuwait	0.95	1.41–2.28	0.27	0.99	0.89–1.11	0.96	1.05	0.95–1.17	0.30	1.12	1.04–1.21	<0.01	1.22	1.11–1.35	<0.01
	Latvia	1.79	0.84–1.22	<0.01	1.64	1.24–2.15	<0.01	0.57	0.49–0.65	<0.01	1.28	1.08–1.52	0.01	0.95	0.83–1.09	0.45
	Lithuania	1.01	0.94–1.18	0.90	0.94	0.75–1.18	0.59	0.52	0.46–0.59	<0.01	1.17	1.02–1.34	0.02	1.05	0.88–1.24	0.61
	Malaysia	1.05	0.99–1.16	0.40	1.08	0.94–1.24	0.28	0.76	0.68–0.85	<0.01	1.31	1.18–1.45	<0.01	1.18	1.04–1.33	0.01
	Mexico	1.07	0.88–1.17	0.06	1.07	0.97–1.18	0.17	1.01	0.92–1.10	0.85	1.09	1.02–1.16	0.01	1.14	1.04–1.23	<0.01
	Morocco	1.01	0.55–0.85	0.87	1.16	0.98–1.37	0.07	0.65	0.58–0.73	<0.01	1.16	1.04–1.30	0.01	1.10	0.95–1.28	0.20
	The Netherlands	0.69	0.28–0.74	<0.01	0.41	0.32–0.54	<0.01	0.68	0.49–0.94	0.02	1.21	1.05–1.39	0.01	1.10	0.92–1.31	0.31
	New Zealand	0.46	1.11–1.35	<0.01	0.71	0.41–1.22	0.22	0.60	0.44–0.80	<0.01	1.40	0.99–1.97	0.06	0.95	0.81–1.11	0.50
	Nigeria	1.23	0.98–1.39	<0.01	1.27	1.14–1.42	<0.01	0.64	0.59–0.71	<0.01	1.23	1.14–1.33	<0.01	1.29	1.16–1.43	<0.01
	Panama	1.17	0.75–0.91	0.08	1.25	1.04–1.51	0.02	0.59	0.52–0.67	<0.01	1.06	0.91–1.22	0.46	1.18	0.99–1.42	0.07
	Philippines	0.83	1.09–1.33	<0.01	1.28	1.14–1.43	<0.01	1.04	0.93–1.16	0.51	1.15	1.07–1.23	<0.01	1.22	1.10–1.34	<0.01
	Poland	1.21	0.77–0.92	<0.01	1.24	1.09–1.40	<0.01	0.98	0.88–1.10	0.77	1.34	1.22–1.46	<0.01	1.68	1.47–1.92	<0.01
	Portugal	0.84	0.71–1.16	<0.01	0.89	0.79–1.0	0.05	0.89	0.79–0.99	0.03	0.97	0.90–1.03	0.32	1.01	0.92–1.09	0.89
	Qatar	0.90	1.05–1.28	0.42	0.89	0.67–1.16	0.38	0.75	0.64–0.87	<0.01	1.57	1.36–1.82	<0.01	1.26	1.00–1.59	0.05
	Romania	1.16	1.01–1.18	<0.01	1.51	1.33–1.71	<0.01	0.79	0.71–0.88	<0.01	1.31	1.19–1.44	<0.01	1.68	1.48–1.91	<0.01
	Russia	1.09	0.65–1.08	0.03	1.25	1.13–1.38	<0.01	0.93	0.85–1.01	0.09	1.29	1.21–1.38	<0.01	1.59	1.46–1.74	<0.01
	Saudi Arabia	0.84	0.59–1.87	0.17	1.12	0.86–1.45	0.39	0.57	0.49–0.65	<0.01	1.35	1.15–1.58	<0.01	1.63	1.30–2.04	<0.01
	Singapore	1.05	0.29–0.91	0.87	1.65	0.86–3.17	0.13	0.52	0.35–0.78	<0.01	0.88	0.50–1.54	0.65	1.40	0.77–2.53	0.27
	Slovenia	0.51	0.85–1.02	0.02	0.54	0.27–1.08	0.08	1.27	0.72–2.23	0.41	0.91	0.67–1.22	0.52	2.46	0.62–9.68	0.20
	South Africa	0.93	0.66–0.79	0.14	0.89	0.78–1.01	0.07	0.92	0.82–1.03	0.16	1.05	0.97–1.14	0.24	1.19	1.07–1.32	<0.01
	Spain	0.72	0.49–0.86	<0.01	0.81	0.72–0.91	<0.01	0.74	0.66–0.83	<0.01	1.11	1.03–1.19	<0.01	1.11	1.00–1.22	0.04
	Sweden	0.65	0.57–1.03	<0.01	0.62	0.45–0.86	<0.01	0.37	0.25–0.54	<0.01	1.18	0.99–1.39	0.05	1.12	0.91–1.36	0.28
	Switzerland	0.76	1.01–1.29	0.08	0.76	0.55–1.04	0.09	0.54	0.45–0.65	<0.01	1.21	1.02–1.43	0.03	1.21	0.99–1.47	0.05
	Taiwan	1.14	0.97–1.15	0.03	1.58	1.38–1.79	<0.01	1.03	0.92–1.16	0.62	1.24	1.15–1.34	<0.01	1.24	1.11–1.37	<0.01
	Thailand	1.06	0.90–1.08	0.22	1.29	1.16–1.43	<0.01	1.01	0.91–1.11	0.85	1.13	1.05–1.22	<0.01	1.13	1.02–1.24	0.02
	Turkey	0.99	1.17–1.46	0.75	0.98	0.88–1.09	0.70	0.96	0.87–1.06	0.39	1.37	1.28–1.47	<0.01	1.53	1.39–1.67	<0.01
	Ukraine	1.29	0.59–0.78	<0.01	1.33	1.16–1.52	<0.01	0.54	0.49–0.59	<0.01	1.40	1.25–1.56	<0.01	2.40	2.04–2.83	<0.01
	United Kingdom	0.68	0.85–1.01	<0.01	0.69	0.59–0.81	<0.01	0.80	0.68–0.93	<0.01	1.00	0.92–1.10	0.93	0.99	0.89–1.11	0.86
	United States	0.93	0.88–1.07	0.09	1.07	0.96–1.19	0.23	0.65	0.59–0.72	<0.01	1.12	1.05–1.20	<0.01	1.27	1.16–1.39	<0.01
	Venezuela	0.97	1.07–1.92	0.56	1.05	0.93–1.19	0.43	1.08	0.96–1.21	0.19	1.05	0.97–1.13	0.25	1.11	1.00–1.22	0.04
Bacteria	*E. coli*	Ref														
	*Citrobacter* spp.	1.43	1.07–1.93	0.02	1.58	1.08–2.31	0.02	2.00	1.58–2.52	<0.01	2.04	1.67–2.49	<0.01	2.79	1.98–3.92	<0.01
	*Enterobacter* spp.	1.66	1.46–1.88	<0.01	2.33	2.05–2.64	<0.01	1.16	1.04–1.28	0.01	3.03	2.73–3.38	<0.01	4.97	4.18–5.91	<0.01
	*Klebsiella* spp.	0.85	0.81–0.90	<0.01	0.73	0.67–0.78	<0.01	1.24	1.17–1.31	<0.01	1.63	1.57–1.69	<0.01	1.72	1.63–1.81	<0.01
	*Morganella* spp.	1.32	0.21–8.26	0.77	0.95	0.05–17.1	0.97	2.60	1.87–3.62	<0.01	2.30	1.86–2.84	<0.01	1.61	0.55–4.71	0.38
	*Proteus* spp.	0.93	0.85–1.03	0.17	0.28	0.24–0.32	<0.01	0.70	0.60–0.81	<0.01	9.98	9.35–10.7	<0.01	2.16	1.99–2.34	<0.01
	*Providencia* spp.	1.91	1.60–2.28	<0.01	1.44	1.17–1.76	<0.01	1.50	1.26–1.77	<0.01	3.55	3.09–4.08	<0.01	3.72	3.08–4.49	<0.01
	*Raoultella* spp.	0.52	0.19–1.35	0.18	0.66	0.14–3.05	0.60	0.91	0.11–7.35	0.93	2.35	1.24–4.43	0.01	1.32	0.78–2.24	0.29
	*Serratia* spp.	0.75	0.61–0.93	0.01	0.23	0.18–0.29	<0.01	0.40	0.32–0.51	<0.01	4.67	4.16–5.26	<0.01	5.39	4.41–6.59	<0.01
Gender	Male	Ref														
	Female	0.98	0.96–1.00	0.08	0.97	0.95–0.99	0.04	1.00	0.98–1.03	0.82	0.99	0.98–1.01	0.59	0.99	0.97–1.02	0.63
Age group	0 to 2 Years	Ref														
	13 to 18 Years	0.96	0.85–1.07	0.42	1.03	0.89–1.18	0.66	0.98	0.88–1.09	0.69	1.00	0.91–1.09	0.94	1.12	0.99–1.27	0.08
	19 to 64 Years	1.07	0.99–1.16	0.08	1.07	0.97–1.18	0.18	1.00	0.91–1.08	0.93	1.08	1.01–1.15	0.02	1.17	1.07–1.28	<0.01
	3 to 12 Years	0.93	0.85–1.01	0.10	1.01	0.89–1.12	0.92	0.98	0.89–1.07	0.63	1.04	0.98–1.11	0.23	1.04	0.95–1.13	0.43
	65 to 84 Years	1.11	1.02–1.19	0.01	1.11	0.99–1.22	0.05	1.03	0.94–1.12	0.50	1.06	0.99–1.13	0.07	1.17	1.07–1.28	<0.01
	85 and over	1.07	0.98–1.16	0.15	1.05	0.93–1.17	0.43	0.99	0.89–1.09	0.86	1.07	0.99–1.14	0.05	1.11	1.01–1.22	0.03
Specialty	Clinic/Office	Ref														
	General Unspecified ICU	1.03	0.96–1.09	0.47	1.07	0.98–1.16	0.10	1.07	0.99–1.16	0.05	1.05	0.99–1.10	0.07	1.05	0.97–1.13	0.21
	Medicine-General	1.01	0.97–1.06	0.48	1.06	1.01–1.11	0.02	1.09	1.04–1.14	<0.01	1.03	0.99–1.06	0.05	1.01	0.97–1.05	0.55
	Medicine-ICU	1.06	1.01–1.11	0.02	1.15	1.08–1.22	<0.01	1.05	1.00–1.11	0.04	1.07	1.03–1.10	<0.01	1.09	1.04–1.15	<0.01
	None Given	1.02	0.94–1.09	0.64	1.05	0.96–1.15	0.24	1.13	1.04–1.23	<0.01	1.00	0.94–1.06	0.90	1.01	0.93–1.09	0.83
	Other	1.01	0.93–1.11	0.75	1.02	0.92–1.13	0.68	1.08	0.98–1.19	0.12	1.00	0.94–1.07	0.95	0.98	0.89–1.07	0.60
	Pediatric-General	1.05	0.96–1.16	0.28	0.96	0.84–1.08	0.46	1.07	0.96–1.18	0.22	1.04	0.97–1.12	0.25	0.99	0.89–1.09	0.86
	Pediatric-ICU	1.06	0.96–1.17	0.22	1.14	1.00–1.28	0.04	1.06	0.95–1.17	0.28	1.03	0.96–1.11	0.38	0.98	0.88–1.09	0.74
	Surgery-General	1.00	0.95–1.05	0.99	1.03	0.97–1.09	0.33	1.03	0.98–1.08	0.25	1.06	1.02–1.09	<0.01	1.05	1.01–1.09	0.03
	Surgery-ICU	1.04	0.98–1.09	0.19	1.07	1.00–1.15	0.04	1.01	0.95–1.07	0.76	1.11	1.06–1.16	<0.01	1.11	1.05–1.18	<0.01
Year	Year	1.05	1.05–1.06	<0.01	0.97	0.96–0.97	<0.01	0.65	0.65–0.66	<0.01	1.03	1.02–1.03	<0.01	1.09	1.09–1.1	<0.01
*bla* _CTX-M-1_	Yes	6.98	6.69–7.29	<0.01	2.97	2.83–3.11	<0.01	3.09	2.96–3.23	<0.01	0.81	0.79–0.83	<0.01	0.87	0.84–0.91	<0.01
*bla* _CTX-M-2_	Yes	5.46	4.96–6.00	<0.01	1.54	1.37–1.72	<0.01	2.70	2.44–2.99	<0.01	1.06	0.98–1.14	0.13	2.46	2.15–2.82	<0.01
*bla* _CTX-M-8/25_	Yes	4.41	3.81–5.11	<0.01	0.34	0.28–0.43	<0.01	3.07	2.62–3.61	<0.01	0.93	0.83–1.05	0.27	0.89	0.75–1.06	0.20
*bla* _CTX-M-9_	Yes	3.98	3.77–4.20	<0.01	0.58	0.55–0.62	<0.01	3.07	2.91–3.25	<0.01	0.84	0.81–0.86	<0.01	0.83	0.79–0.87	<0.01
VEB	Yes	3.24	2.70–3.89	<0.01	6.23	4.90–7.91	<0.01	1.91	1.62–2.25	<0.01	0.64	0.56–0.73	<0.01	0.57	0.47–0.67	<0.01
PER	Yes	2.98	2.09–4.24	<0.01	7.44	4.48–12.3	<0.01	1.26	0.89–1.79	0.19	1.21	0.88–1.65	0.25	0.93	0.64–1.35	0.70
GES	Yes	0.86	0.62–1.19	0.37	2.17	1.56–3.03	<0.01	0.83	0.62–1.11	0.21	1.61	1.22–2.12	<0.01	1.87	1.27–2.76	<0.01
ACC	Yes	0.90	0.62–1.31	0.59	9.62	7.15–12.9	<0.01	2.88	2.18–3.79	<0.01	0.69	0.55–0.86	<0.01	0.68	0.50–0.92	0.01
CMY-2	Yes	0.37	0.33–0.40	<0.01	3.29	3.06–3.54	<0.01	1.06	0.97–1.14	0.18	1.46	1.39–1.53	<0.01	1.09	1.02–1.16	0.01
DHA	Yes	0.33	0.29–0.36	<0.01	2.14	1.98–2.30	<0.01	0.70	0.64–0.76	<0.01	2.20	2.09–2.32	<0.01	1.05	0.99–1.12	0.12
FOX	Yes	0.44	0.18–1.06	0.07	2.12	1.09–4.18	0.03	0.87	0.42–1.81	0.70	0.85	0.61–1.17	0.31	0.87	0.63–1.21	0.41
ACT	Yes	0.62	0.31–1.23	0.17	0.36	0.10–1.28	0.12	0.30	0.05–1.83	0.19	1.77	0.82–3.83	0.14	0.80	0.56–1.13	0.21
KPC	Yes	4.14	3.92–4.37	<0.01	3.44	3.25–3.64	<0.01	1.98	1.88–2.09	<0.01	17.1	16.5–17.7	<0.01	67.38	63.7–71.2	<0.01
OXA	Yes	1.22	1.17–1.26	<0.01	1.23	1.16–1.31	<0.01	0.97	0.93–1.01	0.17	7.46	7.13–7.80	<0.01	21.21	19.8–22.7	<0.01
NDM	Yes	2.09	1.99–2.19	<0.01	2.57	2.45–2.69	<0.01	1.06	1.02–1.10	0.01	10.1	9.59–10.6	<0.01	37.68	34.9–40.6	<0.01
IMP	Yes	3.71	2.91–4.73	<0.01	5.77	4.48–7.42	<0.01	2.37	1.85–3.03	<0.01	5.39	4.22–6.90	<0.01	19.40	13.1–28.7	<0.01
VIM	Yes	3.28	2.89–3.73	<0.01	3.51	3.06–4.02	<0.01	1.58	1.41–1.76	<0.01	7.76	6.83–8.80	<0.01	15.05	12.4–18.2	<0.01
SHV	Yes	1.22	1.16–1.29	<0.01	2.46	2.30–2.64	<0.01	0.84	0.79–0.89	<0.01	0.95	0.91–0.98	0.01	1.05	0.99–1.11	0.05
TEM	Yes	1.09	1.07–1.12	<0.01	1.06	1.02–1.08	<0.01	1.09	1.06–1.11	<0.01	1.02	1.00–1.04	0.01	1.01	0.99–1.04	0.27

MIC: minimum inhibitory concentration; Ref: reference; ICU: intensive care unit.

## Data Availability

The data presented in this study are available on request from the corresponding author.

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
