# Peer review of "Predictive Modeling of Phenotypic Antimicrobial Susceptibility of Selected Beta-Lactam Antimicrobials from Beta-Lactamase Resistance Genes"

_antibiotics, 2024, doi:10.3390/antibiotics13030224_

Round 1
Reviewer 1 Report
Comments and Suggestions for Authors
General comments:
Because an isolate could harbour a resistance gene without expressing it phenotypically (i.e. silent gene), it might be difficult to use genotype in predicting the phenotype. Of concern, it is well established that treating patients with antimicrobial agents merely on the isolates’ genotypes without ascertaining the phenotype by conducting phenotypic susceptibility testing, could potentially result in development of antimicrobial resistance and toxicity. Therefore, there is need for the authors to give the scientific community confidence that this manuscript will enhance diagnosis and management of beta-lactam resistance-associated infections. Further, I expected to see a table summarizing the genotypes and the predicted phenotypes. For example, if blaCTX-M-55 is present in an isolate, the predicted phenotype would be resistance to ceftiofur, ceftriaxone, ceftazidime, imipenem, etc. However, while the spectrum of the beta-lactamases is well known, this study may potentially stimulate further interest in predictive modelling of resistance phenotype using genotype. Finally, it is surprising that blaOXA-48 carbapenemase was not mentioned. Does it imply that it has never been reported in any of the databases consulted in this study.
Specific comments
Abstract
20: “In the presence” – use small letter to begin “in”
23: Begin Carbapenemase with small letter
26: “in retail meat” – begin “in” with capital letter
Introduction
54: change last resort to last-resort, then “AMR bacteria” should be antimicrobial-resistant bacteria. Note that AMR in the manuscript represented antimicrobial resistance; so AMR was used as adjective whereas antimicrobial-resistant is a noun. Therefore, correct it in line 60,661,66,68
68: Something is missing in “…one the most fatal AMR infections”
72: A reader expects to know why “AMR genotype does not always correspond to the AMR phenotype”
104: Begin “Gram” which is name of a person with capital letter in “(gram-positive and gram-negative)”
105: Check the correct spelling of “Intraabdominal”
107-108: National Antimicrobial Resistance Monitoring System for Enteric Bacteria (NARMS)- the acronym should appear after system
110: bacteria genus or bacterial genus? Also check bacteria species in line 378. In line 110, also write E. coli in full first time used
Considering that the problem of antimicrobial resistance is more in resource-limited countries as acknowledged in line 289 than in high-income countries, and that that phenotypic methods are gold standards for clinical management of infections as stated in line 278, the authors should highlight the limitations of molecular techniques and the predictive modeling in the Introduction.
Results
In the prediction of the MIC, the authors used “significantly higher” and “significantly lower” throughout the result section. But, in the methodology, there was no mention of using neither CLSI nor EUCAST MIC breakpoints that are considered the standard. So, I could not understand the criteria used to compare the results before noting they were statistically significant. This needs to be clarified.
Table 1: Some cells had “Ref” but this was not defined underneath the table. And was absence of data in some cases affected the analysis?
227: The authors noted that MIC of cefoxitin increased in the presence of blaTEM-1. Cefoxitin is a cephamycin whose resistance is encoded majorly by AmpC; therefore, using blaTEM-1 to predict cefoxitin phenotype may be erroneous
I think for reproducibility, it is important to mention the tool used in plotting the graphs in figures 1 to 8
Discussion
267: bacterial of? Or members of
270: robust/ weighted least – delete space
302: multi-drug resistance or multidrug resistance
318: Reference 38 is CDC criteria. Does it mean that the criteria were used in classifying the MIC as high or low
325: It is largely known that cefotaxime is more important than ceftriaxone in ESBL assay
Methods
369: What does “on date” mean?
375: Begin animal names with small letter, also in 386
382-383: Something is missing in “information on the MIC cefepime (4th generation)…” Also, generation should be joined to the adjective with a hyphen
412-414: This statements “There was an interaction between blaCMY-2 and year, blaCMY-2 and meat type, blaCMY-2 and genus, genus, and meat type for Cefoxitin in retail meat data. For ceftiofur, there was an interaction between blaCMY-2 and year, blaCMY-2, and meat type.” Were outcomes of the statistical analysis using regression model. I believe it should be in the result section. This statement as well in 416-418 sounded like result: “Heteroscedasticity was detected in the initial model where the variability of the errors is not constant across all levels of the independent variables, leading to a transition to a weighted least squares model (WLS) [43]. It assigned different weights to different observations based on the variance of the errors.”
415: What is the full meaning of “AIC”
Author Response
Thank you for your review, we appreciate your insight, this has helped to improve our manuscript.
General comments:
Because an isolate could harbor a resistance gene without expressing it phenotypically (i.e. silent gene), it might be difficult to use genotype in predicting the phenotype. Of concern, it is well established that treating patients with antimicrobial agents merely on the isolates’ genotypes without ascertaining the phenotype by conducting phenotypic susceptibility testing, could potentially result in the development of antimicrobial resistance and toxicity. Therefore, there is a need for the authors to give the scientific community confidence that this manuscript will enhance the diagnosis and management of beta-lactam resistance-associated infections. Further, I expected to see a table summarizing the genotypes and the predicted phenotypes. For example, if blaCTX-M-55 is present in an isolate, the predicted phenotype would be resistance to ceftiofur, ceftriaxone, ceftazidime, imipenem, etc. However, while the spectrum of the beta-lactamases is well known, this study may potentially stimulate further interest in the predictive modelling of resistance phenotype using genotype. Finally, it is surprising that blaOXA-48 carbapenemase was not mentioned. Does it imply that it has never been reported in any of the databases consulted in this study?
Response:
We sincerely appreciate your thoughtful and constructive feedback on our manuscript titled "Predictive modeling of phenotypic antimicrobial susceptibility of selected beta-lactam antimicrobials from beta-lactamase resistance genes”. Your insights have been invaluable in enhancing the overall quality and scientific rigor of our work. Below, we address each of your comments and provide clarification and improvements accordingly.
Our main emphasis was on predicting MIC values of antimicrobials using the genotypic information available in the dataset. It is crucial to clarify that our objective did not include predicting phenotypes from particular genes. In terms of data representation, we opted for the ATLAS dataset for human samples and the NARMS retail meat dataset of food animals. While we included all accessible beta-lactamase genes in our analysis, we did not investigate blaOXA-48 carbapenemase in this project. We currently have another project under analysis that is specifically for specific alleles of beta-lactamase epidemiology. Our focus was to look at the beta-lactamase gene family.
Specific comments:
Abstract
20: “In the presence” – use the small letter to begin “in”
Response: Thank you. Corrected accordingly in line number 20.
23: Begin Carbapenemase with a small letter
Response: Thank you. Corrected accordingly in line number 23.
26: “In retail meat” – begin “in” with a capital letter
Response: Thank you. Corrected accordingly in line number 26.
Introduction
54: change last resort to last-resort, then “AMR bacteria” should be antimicrobial-resistant bacteria. Note that AMR in the manuscript represented antimicrobial resistance; so AMR was used as an adjective whereas antimicrobial-resistant is a noun. Therefore, correct it in line 60,61,66,68
Response: Thank you for your suggestions. Corrected accordingly in line number 54-70.
68: Something is missing in “…one the most fatal AMR infections”
Response: Thank you. Corrected accordingly in line number 69-70.
72: A reader expects to know why “AMR genotype does not always correspond to the AMR phenotype”
Response: Thank you for your suggestions. We updated the statement in line number 72-77.
104: Begin “Gram” which is the name of a person with a capital letter in “(gram-positive and gram-negative)”
Response: Thank you. Corrected accordingly in line number 107-108.
105: Check the correct spelling of “Intraabdominal”
Response: Thank you. Corrected the spelling in line number 108.
107-108: National Antimicrobial Resistance Monitoring System for Enteric Bacteria (NARMS)- the acronym should appear after system
Response: Thank you. Corrected accordingly in line number 110-111.
110: bacteria genus or bacterial genus? Also, check bacteria species in line 378. In line 110, also write E. coli in full first time used
Response: Thank you. Corrected accordingly in line number 113 and 411.
Considering that the problem of antimicrobial resistance is more in resource-limited countries as acknowledged in line 289 than in high-income countries, and that phenotypic methods are gold standards for clinical management of infections as stated in line 278, the authors should highlight the limitations of molecular techniques and the predictive modeling in the Introduction.
Response: Thank you for your suggestions. We have added the limitation of molecular techniques in line number 72-77.
Results
In the prediction of the MIC, the authors used “significantly higher” and “significantly lower” throughout the result section. However, in the methodology, there was no mention of using either CLSI or EUCAST MIC breakpoints which are considered the standard. So, I could not understand the criteria used to compare the results before noting they were statistically significant. This needs to be clarified.
Response: Thank you for your query. In the case of ATLAS, MICs were established through broth microdilution by CLSI guidelines, and the interpretation utilized the 2020 CLSI breakpoints. For NARMS, MIC breakpoints followed the guidelines established by the CDC NARMS. Please see the line number (419-423)
Table 1: Some cells had “Ref” but this was not defined underneath the table. And was the absence of data in some cases affected the analysis?
Response: In our analysis, we utilized a multivariable linear regression model with a designated "Ref" cell serving as the “reference” or comparative group consistent with regression modeling. The analysis involved comparing other categories of a specific variable about this reference category. For example, if age 0-2 years is the reference, we can compare adult vs 0-2 years, 5-10 years vs 0-2 years, etc. In this case, 0-2 years group is the reference for comparison. It's important to note that the data was not absent and had no impact on the conducted analysis.
227: The authors noted that the MIC of cefoxitin increased in the presence of blaTEM-1. Cefoxitin is a cephamycin whose resistance is encoded majorly by AmpC; therefore, using blaTEM-1 to predict cefoxitin phenotype may be erroneous.
Response: Thank you for your concern. Based on the dataset, our analysis revealed a significant increase in cefoxitin MIC in the presence of AmpC, specifically the blaCMY-2 gene. Over the 17 years examined, we observed an average MIC increase of 8.66 μg/mL associated with the presence of blaCMY-2. This substantial increase contrasts with the minimal impact observed when using blaTEM-1, where the average MIC increase was only 1.04 μg/mL. The use of blaTEM-1 alone to predict the cefoxitin phenotype may not be adequate, however, this is what the predictive model produced. We acknowledge the complexity of cefoxitin resistance and the limitations of relying solely on blaTEM-1. The model showed that blaCMY-2 increases the MIC of cefoxitin better than blaTEM-1 which is accurate, however, blaTEM-1 showed poor hydrolysis of cefoxitin which is consistent with what is expected.
I think for reproducibility, it is important to mention the tool used in plotting the graphs in Figures 1 to 8
Response: Thank you for your concern. The models were visualized in plots using the R package sjPlot. We have updated the revised manuscript in line number 468-469.
Discussion
267: bacterial of? Or members of
Response: Thank you. Corrected accordingly in line number 300.
270: robust/ weighted least – delete space
Response: Thank you. Corrected accordingly in line number 303.
302: multi-drug resistance or multidrug resistance
Response: Thank you. Corrected accordingly in line number 335.
318: Reference 38 is CDC criteria. Does it mean that the criteria were used in classifying the MIC as high or low
Response: Thank you for bringing up this concern. The MIC breakpoints for NARMS adhered to the guidelines set forth by the CDC NARMS. I have incorporated this information into the methodology section of the revised manuscript, specifically in lines 422-423
325: It is largely known that cefotaxime is more important than ceftriaxone in ESBL assay
Response: Thank you. We have corrected the statement in line number 357-358.
Methods
369: What does “on date” mean?
Response: Thank you for noting that error. The ATLAS dataset, was acquired through Vivli - Center for Global Clinical Research Data on June 15, 2023. Please see line number 399-402.
375: Begin animal names with small letters, also in 386
Response: Thank you. Corrected accordingly in line numbers 409-410 and 425.
382-383: Something is missing in “information on the MIC cefepime (4th generation)…” Also, generation should be joined to the adjective with a hyphen
Response: Thank you. Corrected accordingly in line number 417-418.
412-414: This statement “There was an interaction between blaCMY-2 and year, blaCMY-2 and meat type, blaCMY-2 and genus, genus, and meat type for Cefoxitin in retail meat data. For ceftiofur, there was an interaction between blaCMY-2 and year, blaCMY-2, and meat type.” Were outcomes of the statistical analysis using a regression model? I believe it should be in the result section. This statement as well as in 416-418 sounded like the result: “Heteroscedasticity was detected in the initial model where the variability of the errors is not constant across all levels of the independent variables, leading to a transition to a weighted least squares model (WLS) [43]. It assigned different weights to different observations based on the variance of the errors.”
Response: We appreciate you bringing up this concern. The result section has been revised accordingly.
415: What is the full meaning of “AIC”
Response: Thank you. The full form of AIC is “Akaike information criterion” We have added the full form in line number 452.
Reviewer 2 Report
Comments and Suggestions for Authors
For my part, I think that only 2 aspects could be improved a little:
- Write the introduction a little more concise
- The texts in the figures must be larger.
Author Response
Reviewer 2:
For my part, I think that only 2 aspects could be improved a little:
- Write the introduction a little more concise
Response: Thank you for your feedback. We have revised the introduction based on your and other reviewers' comments, aiming for a more concise presentation. We appreciate your guidance in refining this section.
- The texts in the figures must be larger.
Response: Thank you for highlighting this concern. In response to your comment, we have increased the font size for all figure texts in the revised manuscript.
Reviewer 3 Report
Comments and Suggestions for Authors
A very interesting manuscript evaluating predictive modeling of phenotypic antimicrobial susceptibility of selected beta-lactam antimicrobials from beta-lactamase resistance genes is presented.
The article provides a new perspective on AMR assessment, especially in terms of the time requirements that standard phenotypic tests require to determine bacterial susceptibility/resistance to antibiotics.
I have minor comments on the text:
In Table 1 and at the same time throughout the text, I recommend giving the full names of bacteria (e.g. Klebsiella spp). In the current version, the names of the bacteria are not uniform, e.g. Klebsiella spp and Klebsiella are listed.
On line 313 it says...in the presence of Carbapenemase... there is no reason for the capital "C", it should properly be ..in the presence of carbapenemase...
Author Response
Reviewer 3:
A very interesting manuscript evaluating predictive modeling of phenotypic antimicrobial susceptibility of selected beta-lactam antimicrobials from beta-lactamase resistance genes is presented.
The article provides a new perspective on AMR assessment, especially in terms of the time requirements that standard phenotypic tests require to determine bacterial susceptibility/resistance to antibiotics.
Response: Thank you for your positive feedback on our manuscript. We appreciate your recognition of the novel perspective our article brings, particularly in addressing the time requirements associated with standard phenotypic tests for determining bacterial susceptibility/resistance to antibiotics.
I have minor comments on the text:
In Table 1 and at the same time throughout the text, I recommend giving the full names of bacteria (e.g. Klebsiella spp). In the current version, the names of the bacteria are not uniform, e.g. Klebsiella spp and Klebsiella are listed.
Response: Thank you for your suggestions. In the revised version, we have ensured consistency by providing the full names of bacteria throughout the text and table.
On line 313 it says...in the presence of Carbapenemase... there is no reason for the capital "C", it should properly be ..in the presence of carbapenemase...
Response: Thank you for your suggestions. I have fixed the carbapenemase word throughout the revised manuscript.
Reviewer 4 Report
Comments and Suggestions for Authors
The authors have conducted an extensive study on antimicrobial resistance in Enterobacteriaceae. The objective of the work was to develop a predictive model to estimate susceptibility to beta-lactams using data from bacterial antimicrobial resistance genes. The authors completed the study through a statistical analysis that included factors such as type of microorganism, anatomical origin of the isolate, and place and period of infection. Likewise, they included information on antimicrobial resistance of animal origin.
The study is very complete, highlighting the importance of surveillance and control of antimicrobial resistance. As observations, I suggest including the type of study design, the inclusion and exclusion criteria of the beta-lactamase genes, and the selection criteria for the databases from which the study data were extracted.
Author Response
Reviewer 4:
The authors have conducted an extensive study on antimicrobial resistance in Enterobacteriaceae. The objective of the work was to develop a predictive model to estimate susceptibility to beta-lactams using data from bacterial antimicrobial resistance genes. The authors completed the study through a statistical analysis that included factors such as type of microorganism, anatomical origin of the isolate, and place and period of infection. Likewise, they included information on antimicrobial resistance of animal origin.
The study is very complete, highlighting the importance of surveillance and control of antimicrobial resistance. As observations, I suggest including the type of study design, the inclusion and exclusion criteria of the beta-lactamase genes, and the selection criteria for the databases from which the study data were extracted.
Response: Thank you for your insightful observations and positive feedback regarding our study. We value your suggestions aimed at improving the comprehensiveness of our manuscript. In response, we have refined our methodology by adopting a cross-sectional study design for this research. We ensured the inclusion of all available beta-lactamase genes in our analysis. Our data sources were two prominent AMR surveillance programs, ATLAS and NARMS. ATLAS contributed data from human samples, while NARMS provided data on retail meat from food animals. This dual dataset selection was deliberate to comprehensively represent both human and animal perspectives, adding depth and robustness to our study.